# Glycan Nanobiosensors

**DOI:** 10.3390/nano10071406

**Published:** 2020-07-19

**Authors:** Filip Kveton, Anna Blsakova, Peter Kasak, Jan Tkac

**Affiliations:** 1Institute of Chemistry, Slovak Academy of Sciences, Dubravska cesta 9, 845 38 Bratislava, Slovakia; Filip.Kveton@savba.sk (F.K.); Anna.Blsakova@savba.sk (A.B.); 2Center for Advanced Materials, Qatar University, Doha 2713, Qatar

**Keywords:** glycans, biosensors, nanomaterials, nanoparticles

## Abstract

This review paper comprehensively summarizes advances made in the design of glycan nanobiosensors using diverse forms of nanomaterials. In particular, the paper covers the application of gold nanoparticles, quantum dots, magnetic nanoparticles, carbon nanoparticles, hybrid types of nanoparticles, proteins as nanoscaffolds and various nanoscale-based approaches to designing such nanoscale probes. The article covers innovative immobilization strategies for the conjugation of glycans on nanoparticles. Summaries of the detection schemes applied, the analytes detected and the key operational characteristics of such nanobiosensors are provided in the form of tables for each particular type of nanomaterial.

## 1. Glycomics

Carbohydrates—along with lipids, nucleic acids and proteins—are representatives of the biomolecules essential for life with glycans (complex carbohydrates) densely covering the cellular surface with involvement in numerous processes [1,2]. A glycome is a complete collection of all the glycans present in cells, tissues or organisms at any particular time [3]. Glycosidic bonds between two carbohydrate-building blocks are created by the coupling of an anomeric hydroxyl group of one carbohydrate with any of the hydroxyl groups of the second one. Further diversity can be achieved by way of mutual combinations of these carbohydrates, via a variety of bonds (1–3; 2–6; etc.), using different anomeric states (α vs. β), branching, length and substituted components (phosphate, sulfate, etc.) [4]. The theoretical number of hexa-saccharide glycans is remarkable and several orders of magnitude larger (1.9 × 10^11^) than hexapeptides (6.4 × 10^7^) or hexanucleotides (4096 combinations) [5]. Oligosaccharides are able to bind more strongly with different glycan-binding proteins than monosaccharides, thanks to the variety in structures and conformations [6]. For example lectins bind to monosaccharides with affinity constant in the millimolar range [2]. Oligosaccharides bind to lectins with affinity constant in the micromolar range despite the fact that oligosaccharides can bind to lectins via multiple contacts [1]. This is due to absence of a deeper binding pocket on the surface of lectins allowing competitive solvent interactions [1]. On the other hand, interaction of lectins with monosaccharides can be enhanced in cases lectins are assembled from homo-oligomeric subunits and each subunit interacts with a monosaccharide [1].

Glycosylation—a sophisticated form of a post-translational modification—is responsible for a proper glycan anchoring on biomolecule scaffolds by a step-by-step enzymatic addition of carbohydrate chains [4,7]. In proteins, glycans can be attached to at least nine out of 20 types of amino acid residues. Two of the most dominant processes include *N*-glycosylation (amide linkages to asparagine residues) and *O*-glycosylation (glycosidic bond to serine or threonine). It is notable that these reactions influence more than 50% of all proteins in organisms and nearly all surface proteins of cells (receptors or adhesion proteins). Glycosylation takes place in the endoplasmic reticulum, but more predominantly in the Golgi apparatus and the process is not template-driven. Instead, glycans are synthesized by the action of a series of enzymes [1]. The result of the reaction is a vast diversity of glycan structures with many important cellular functions [8,9].

For many years, the role of sugars was considered to be primarily nutritional, but the role of glycans becomes progressively more complex [9]. Conjugates are part of interfacial layers of cells and responsible for mediation of the first contact in the host-pathogen interactions [10,11]. Highly specific, but weak protein–glycan interactions are commonly observed in nature and have an essential role in many cellular mechanisms, e.g., cell–cell and cell–biomolecule interactions, stabilization of tertiary structure of proteins, mechanism of signaling molecules or disease progression and infection of pathogens including toxins, bacteria and viruses [8,10,11,12,13]. As an illustration, sialic acid (SA) is used as a viral receptor molecule by a host cell. Binding of a particular virus to the cell surface is based on interactions between the viral glycoprotein—hemagglutinin (HA) and cell-surface glycans terminated in SA. Ordinarily, human viral strains are linked to α–2, 6–SA moieties, whereas avian viruses predominantly bind to α–2,3–SA structures [14] (Figure 1).

Biologic changes in the organism can result in the modification of glycans. For example, malignant cells undergo significant alterations in terms of glycan expression. Aberrant glycosylation (predominantly characteristic for cancer development) causes differences in the molecular patterns of healthy individuals and indicates the pathophysiological state of biologic systems [12,15,16]. Hence, the identification of specific glycans is a crucial step in disease diagnostics, the prognosis or monitoring of various diseases [15,16] and, thanks to glycans, new therapeutic strategies can be developed for major diseases [9]. There are several factors responsible for glycan recognition, including glycan-to-glycan spacing on the interface and length of a linker via which the glycan is attached to the interface [17].

## 2. Glycan-Functionalized Nanoparticles (NPs)

Ligand-modified surfaces are probably enhanced due to multivalent interactions between surface-anchored clusters composed of many saccharide units. This phenomenon is denoted as a glycocluster effect and is able to multiply avidity and activity in biologic systems [19]. On the other hand, the mechanical properties and mobility of a ligand can affect the biorecognition process to a great extent. Glycan-functionalized nanoparticles (NPs) consist of two main domains, the nanoparticle core with the shell formed by the glycan structures (mono-, oligo- or polysaccharides). The combination of these two domains is a crucial step in the application and effectiveness of glycan-NPs.

Some of the biologic and medical problems could be solved by the aid of glyconanotechnology. This scientific discipline represents a synergy between nanotechnology and glycomics, both of which are emerging technologies [9]. Nanotechnology is a general term for a scientific discipline dealing with the creation and/or use of materials having at least one dimension within the range of one to one-hundred nanometers [9]. Here we use a term nanomaterial as a material, which consists of structured components (nanoparticles) with at least one dimension less than 100 nm [20]. Life systems inspire humanity to apply nanotechnology to solve biologic and medical problems and to construct new nanomaterials and devices [9]. Nowadays, NPs (Figure 2) are well established in different fields of everyday human life [21], for example, in electronic devices (amplification of immunosensors’ sensitivity) [22,23,24], medical applications (as imaging tools, drug delivery carriers, nanosensors and gene-delivery therapy [9,25,26]), the cosmetics [27] and food industries [28]. Further, NPs with engineered surface characteristics may transfer through the cell membranes without causing their perturbations, thereby allowing direct interactions with cell components [29].

It is challenging to compare the toxicity of nanomaterials with their macroscale counterparts. Current toxicity assays are applied equally both to NPs and to conventional agents. Hence, the evaluation of NPs’ toxicity by current methods may not be sufficient. This is due to increased surface area of NPs strongly adsorbing active agents applied for traditional toxicology assays; optical properties of NPs, which may interfere with fluorescence or optical detection systems; and magnetic properties of NPs, which may interfere with methods based on redox reactions [30]. As a result of such properties of NPs, the methods used for traditional toxicology studies cannot be directly applied for examination of toxicity of NPs [30]. Moreover, NPs can be accumulated in various organs for longer periods than traditional pharmaceutical agents generating oxidative stress, inflammation, cell death and agglomerates within vessels [31]. There is a need for the development of new assays consisting of several approaches [31] including assessment of toxicity of NPs by aquatic organisms [32], which will take into consideration factors such as size, shape, surface area, surface charge, porosity or hydrophobicity which influence the functions and toxicity of nanomaterials [33,34]. It has been established that NPs are able to activate an immune response and induce phagocytic cells that will eliminate them from the bloodstream or may induce immunostimulation which may promote inflammatory disorders or even immunosuppression which increases the host’s susceptibility to infections and cancer. It is generally believed that the integration of NPs with glycans will not elicit a strong immune response increasing the circulation of NPs within the bloodstream [9]. The toxicity of silver NPs functionalized with galactose and mannose moieties to neuronal-like and hepatic cells was lower than in these coated with glucose [35] so it is crucial to finely tune the interfacial glycan composition of NPs in order to achieve the desired function of NPs [36].

Nanoparticles and nanomaterials have unique physical, optical, electrochemical and chemical properties which open up new possibilities when integrated into numerous applications. One of the most interesting things in the nanoworld is the huge surface-to-volume ratio of particles, significantly affecting their physicochemical properties [9,38]. The combination of nanoscale materials with biosensors suggests unique opportunities for the ultrasensitive detection of biomolecules. In this regard, nanomaterials could be used as a possible platform for displaying glycans that allow precise surface positioning and a higher density than in traditional approaches. Additionally, the use of nanomaterials such as carbon nanotubes (CNTs), graphene, magnetic NPs or gold nanoparticles (AuNPs) may enhance the operational characteristics of biosensors [39,40] and other bioanalytical devices [41]. As a consequence of their configuration and exceptional, unique and impressive physicochemical properties, NP-based materials are frequently used in biosensing. AuNPs are quite frequently applied for construction of biosensors since such nanoparticles allow modifying interface in a convenient way using formation of self-assembled monolayers (SAMs) using thiolated biomolecules [42]. A combination of two thiols can effectively tune density of functional groups deposited on the surface and glycan nanobiosensors on AuNPs can be directly formed using thiolated glycans [43]. Although formation of SAM on AuNPs is straightforward with many beneficial properties, it is very important to point out to the fact that components attached to AuNPs via SAM are highly mobile with a possibility that glycan will form clusters over time even though glycans were initially homogeneously distributed over the interface [43]. AuNPs are frequently applied to design electrochemical (modification of the electrodes) and optical (signal nanoprobes) biosensor devices. Quantum dots (QDs), which are usually terminated in hydroxyl groups can be quite easily modified by silane chemistry with final formation of a strong irreversible bond. The initial step is hydrolysis of silane with subsequent condensation reaction creating Si–O bond [43]. Similar to thiolated SAMs, silane chemistry allows forming a monolayer with a possibility to deliver various functional groups, but the process is more difficult to control [43]. QDs are especially applied obviously in combination with optical detection, but also electrochemical detection platform is frequently applied since heavy metals of QDs can be effectively detected by electrochemical means. QDs are mainly applied as signal nanoprobes. Magnetic particles are usually made of Fe_3_O_4_ meaning that interfacial hydroxyl group can be again applied for modification via silane chemistry discussed above for QDs. The obvious advantage of using magnetic particles is to apply them for preconcentration/enrichment of the analyte from complex samples [9]. Magnetic particles can be loaded with various labels to form signal nanoprobes. Carbon nanoparticles can be modified via various routes. For example, planar forms of carbon nanoparticles such graphene or carbon nanotubes can be modified by non-covalent π-π stacking interactions, by covalent grafting of molecules to modified/oxidized carbon nanoparticles or via electrochemically triggered grafting of molecules having diazonium moieties. Carbon nanoparticles are frequently applied for modification of the electrodes or as signal nanoprobes for electrochemical biosensing or can be applied for fluorescent biosensing with carbon nanoparticles effectively applied for fluorescence quenching. Bioconjugation protocols applicable for glycan immobilization are in details discussed in our book chapter [43].

Glycans immobilized on biosensing surfaces could effectively mimic the functional role of cell surface glycans, which is why, by designing glycan-modified nano-interfaces, it was possible to detect glycan-binding biomolecules in a way that resembles natural recognition. Accordingly, a variety of ways of using (bio)nanotechnology in combination with glycans for the analysis of various biomolecules, microbes and viruses are discussed below. Depending on their application, it is of the utmost importance to choose a proper method for the attachment of glycans to NPs. This review does not extend to coverage of the formation and modification of self-assembled monolayers on NPs and bioconjugation techniques for glycan immobilization and controlling interfacial glycan density. The reader is directed towards an excellent review study [44] and a book chapter [43].

However, some immobilization strategies are included here since they facilitate the preparation of switchable interfaces. The beauty of switching approaches described below is regeneration of the interfaces by external trigger in this particular case by light. More information about switchable materials can be found elsewhere [45]. In the first example, a mixed SAM was formed on gold with functional termination containing azo groups [46]. In the azo groups, β-cyclodextrin conjugated with a biocidal quaternary ammonium salt group was incorporated. Illumination by UV light resulted in the switch of the azo group from trans conformation to the cis form releasing β-cyclodextrin conjugate with its cargo (in this case, dead bacterial cells). Upon the switch to visible light, the interface was regenerated [46]. The same approach using azo groups as guest ligands for β-cyclodextrin was applied to photo-reversible capture and the release of bacteria when, instead of a biocidal derivative, mannose units were attached to β-cyclodextrin [47]. Again, by switching to UV light, bacteria were released from the interface (Figure 3) and by switching to visible light the interface was ready to host another β-cyclodextrin with covalently attached mannose units [47]. Another switching interface was constructed using β-cyclodextrin attached to gold surface-confined phenylboronic acid via the interaction of boronic moieties with secondary –OH groups on the rim of β-cyclodextrin. Upon the addition of a cis diol-containing molecule such as fructose, β-cyclodextrin was detached from the phenylboronic acid-modified interface. The molecule of β-cyclodextrin bearing various functional groups was applied to the capture of a protein or the capture and killing of bacteria in a switchable manner [48]. More details on β-cyclodextrin interfaces with a photo- but also redox-switchable modalities can be found in a review study [49]. Another interesting approach to the immobilization of glycans onto carbon nanotubes was based on the radical addition of aryl diazonium species generated thermally in-situ [50]. An alternative to covalent grafting of aryl diazonium species is a spontaneous grafting in case a nanomaterial contains a free electron cloud (plasmons) as in the case of hybrid magnetic particles with a gold shell [51,52] or MXene 2D nanomaterial (a novel form of a 2D hydrophilic nanomaterial made of alternating layers of two elements such as titanium and carbon, i.e., Ti_3_C_2_) [53]. The supramolecular self-organization of diacetylenic-based neoglycolipids into a highly organized ring around single-walled carbon nanotubes (SWCNTs) was recently described with a final formation of a “shish-kebab-like” morphology (Figure 4) [54]. TEM images confirmed the successful binding of lectins to such glyconanorings present on SWCNTs [54].

Another interesting approach, developed for the immobilization of glycans, relies on a low-cost laser-based printing setup (Figure 5) [55]. The technology facilitates the spot-wise patterning of surfaces with defined polymer nanolayers. Such nanolayers function as a reservoir containing different chemicals, chemical building blocks, materials or precursors, all of which can be stacked on top of each other. Upon laser irradiation, polymer-embedded molecules are released and can be deposited on another surface with the thickness of printed nanolayers of 10 nm. The technology also makes possible the mixing of different molecules on the surface (Figure 5C). The proof of concept was also tested by the binding of Con A to a mannose-patterned interface [55].

Several review papers discuss the preparation and application of glycan nanobiosensors to the detection of various glycan-binding biomolecules [35,56,57,58], with only one comprehensively summarizing the literature published in 2016 [44]. Hence, this review study seeks to provide an update in this fascinating scientific field since 2016.

### 2.1. Gold Nanoparticles (AuNPs)

AuNPs are among the most frequent NPs applied in life sciences, due to their compatibility with a thiol-based self-assembly process, high stability and unique optical and electrochemical properties [42,59]. Moreover, AuNPs can be synthesized in a simple way, with interfacial functionalization achieved during a synthetic step [60]. Biocompatible AuNPs covered with thiol-terminated glycans were extensively studied for interactions with glycan-binding affinity probes relying on multivalent glycan presentation. It was shown that these glyco-NPs are able to selectively target specific receptors [61]. They have also been examined in glycan biosensors [62,63,64], in vivo cell imaging and targeting [65], vaccine vectors development [66,67], virus detection [68,69] and drug delivery [70]. Glyco-NPs benefit from having a large geometric surface-to-volume ratio, which means that just one NP is able to offer numerous carbohydrate moieties for the enhancement of biomolecular interactions [71]. In the following section we are discussing AuNPs with the size above the size of nanodots, i.e., 10 nm [72] and mainly without nanotextured surface, which is typical for mesoparticles [73].

The aggregation of AuNPs results in a pronounced color change from red to blue (shift to longer wavelengths) dependent on inter-particle distances (Figure 6) [63]. Color changes of NPs are closely correlated with the resonance between an oscillation of electrons and the incident electromagnetic radiation [74]. It is worth mentioning that optical properties are changed with size for other types of NPs such as quantum dots [75], as well. The AuNPs-based colorimetric assays to study biomolecular interactions with proteins, peptides and DNA, exploit this phenomenon. The optical properties are influenced by the size and shape of AuNPs, the degree of aggregation and the dispersing solvent [63]. Light scattered by a single AuNP can be around one million times more intense than the fluorescence emitted by a dye [76,77]. Langer et al. [78] studied AuNPs of various sizes (30, 42 and 60 nm) for the determination of galectin 9. They noted a relation between the NPs’ size and their colloidal stability and discovered that AuNPs with a size of 60 nm were prone to aggregation, while the most aggregation-resistant were AuNPs with a diameter of 30 nm. Finally, the nanosensor was successfully applied to the detection of galectin 9 down to a level of 1.2 nM using surface-enhanced Raman scattering [78].

The distance-dependent optical property of AuNPs was used as an emerging technique for the simple analysis of influenza viruses resulting in detectable color and plasmon resonance shifts. Zheng et al. [69] reported progress in an application of glycan (via Au–S bond)-modified AuNPs (13 nm) used in a single-step colorimetric process for the detection of 14 major serotypes of viral strains. Interactions between viruses and glycans were analyzed with 7 different SA residues (3 derivatives of α-2,6-linked SA and 4 derivatives of α-2,3-linked SA) by an ELISA method and using a shift in the position of a plasmon band (expressed as A_680_/A_522_). All influenza strains were distinguished using the Au probes. The device based on the shift of a plasmonic band offered a higher LOD when compared to glycan–ELISA (8 HA titer vs. <2 HA titer, respectively) but, on the other hand, a much shorter reaction time (90 min vs. 18+ hours) and simplicity [69]. AuNPs modified with fetuin (a protein containing SA) were used in the electrochemical detection of influenza strain H9N2 [80]. Besides fetuin-modified AuNPs (14 nm), magnetic beads were modified by an anti-M2 antibody recognizing the influenza virus. In the presence of viral particles, a sandwich configuration was formed, and, by a magnetic field, such a complex was transferred onto a screen-printed electrode (SPE) made of carbon. The electrochemical readout was achieved by registering the chronoamperometric signal at −1.00 V. The device was able to detect viral particles with LOD of 8 HAU [80]. Older papers showing the application of plasmonic glycan-based devices based on AuNPs to the detection of lectins are summarized elsewhere [81].

The second equally important factor is the shape of the prepared NPs. Toraskar et al. [82] compared glycodendrons (α-D-mannose- and β-D-galactose-linked) displayed and anchored on spherical and rod-shaped AuNPs. They concluded that the bacterial infection of HeLa cells was better inhibited by the homogeneous glycodendron-conjugated rod-shaped AuNPs than by the heterogeneous ones [82]. Three differently shaped AuNPs (sphere, rod and star-like) were designed by Kikkeri and coworkers [83]. Galactose and mannose moieties were immobilized on AuNPs and used for the quantification and detection of a binding affinity to bacteria *E. coli*. Each shape delivered a different bacterial adhesion value. Nanorods have LOD of bacteria of about 0.03 ± 0.01 μg·mL^−1^, which was 80-fold lower (more sensitive) than for the spherical or star-shaped AuNPs. This was associated with variability in the relative number of mannose molecules involved in the NPs-bacteria interactions. The same group obtained similar results (higher responses of rod-modified NPs than in star and spherical particles) in in vitro experiments with mammalian cells [84].

Otten et al. simulated the biologic heterogeneity of glycans (11 combinations of galactosamine and mannosamine with galactosamine) [85]. Mixed glycan monolayers were immobilized on 60 nm AuNPs. Glycosylated AuNPs were produced with the possibility of changing color from red to blue in the UV-Vis spectra as a result of an interaction with carbohydrate-binding proteins. In the study, three different lectins were investigated and their binding to glyco AuNPs was analyzed using a linear discrimination analysis with separation into distinct clusters. The glyconanobiosensor could detect lectins down to a few nM concentration [85].

Selvaprakash’s team prepared AuNPs encapsulated in ovalbumin (attachment via cysteines residues) with the surface decorated with hybrid mannose, glucose and Galβ(1→4)GlcNAc-terminated glycan. This was the first study involving an application of AuNPs as affinity probes for the analysis of multiple lectins. NPs were produced in a one-pot reaction of chicken egg white proteins with an aqueous tetrachloroaurate solution [86].

Glyco AuNPs of 20 nm in size were prepared by a one-pot synthesis of AuNPs from gold salt in the presence of maltose [87]. These glycan-modified AuNPs were then used in the detection of Con A down to 23 pM with an affinity constant determined as 67 nM. The selectivity of the device was investigated by measuring three other lectins with a negligible shift of the plasmonic band. In addition, a combination of Con A with glycan AuNPs probe was used for cancer cell imaging [87].

Won et al. presented a switchable interface prepared by modifying AuNPs with polymeric “gates”, which either allow the lectin to bind to the glycan present on AuNPs or not [88]. Two different polymer chains were immobilized on the AuNPs with a size of 60 nm, a shorter one with an attached glycan and a longer one acting as an active gate. Under critical temperature, the longer polymer chain had an extended conformation with its shrinkage upon a shift of temperature above the critical value of 40 °C, exposing underlying glycans (Figure 7). Lectins were detected with LOD down to μg·mL^−1^ [88]. In the next study from the same group, AuNPs were rendered responsive towards lectins and Ca^2+^ ions [89]. A similar strategy using microgels instead of AuNPs was used in the detection of Con A and *E. coli*, which could be separated after binding by filtration [19,90].

Many pathogen-related diseases are caused by food–borne bacterial strains. Proteins on the bacterial surface are able to bind to different types of glycans on the host cells. Kaushal et al. focused on a rapid detection of the food–borne pathogen *E. coli* with the application of gold nanorods-based sensors with good near-infrared absorption and scattering in surface plasmon resonance wavelength regions [91]. Polyethylene glycol-coated Au nanorods were immobilized by amine-terminated sugars—4-aminophenyl α-D-mannopyranoside and 4-aminophenyl β-D-galactopyranoside. The study showed that nanobiosensors could detect lectins down to M concentration range. *E. coli* bacteria bind to mannose-modified AuNPs and *Pseudomonas aeruginosa* was used as a control to check the aggregation of the galactose-modified nanorods. The assay was performed in a 96-well plate for a visible colorimetric detection [91].

A direct comparison of rod-shaped and spherical AuNPs modified by glycans to detect two different *E. coli* strains indicated that the LOD was significantly lower using rod-shaped NPs (20 μg·mL^−1^) than with spherical AuNPs (200 μg·mL^−1^) [82].

A plasmonic field-effect transistor (FET)-based device was also used for the detection of lectins in the concentration window from 10^−10^ to 10^−4^ M [92]. To fabricate the device, AuNPs were deposited on an InGaZnO gate with carbohydrate immobilized on AuNPs using UV irradiation (Figure 8). A transducing mechanism relied on converting hot plasmonic electrons into an electric current measured by the device [92].

A good overview of various routes for the modifications of AuNPs, which could then be applicable to glycan immobilization, is shown in two review papers [65,93].

Silver-coated gold nanorods were used [77] as scaffolds for mannose immobilization and used as a study model for the detection, killing and quantification of the interaction between glyco-AuNPs and bacteria *E. coli*. When localized surface plasmon resonance was used in the detection of Con A, the lectin could be detected down to a low nM concentration [77].

Silver on AuNPs were prepared from 2–3 nm AuNPs by growing them to a size of 30 nm and then the AuNPs were covered in a silver shell to produce AgAu nanocubes with the size of 55 nm [94]. Such nanoparticles were immobilized on an ITO slide and then modified by thiolated mannose. Localized surface plasmon resonance (LSPR) was used for measurement of the changes in the red shift in the presence of Con A. The authors claim that the binding of Con A to a single AgAu nanocube was feasible with LOD down to a few nM for Con A. Three other lectins, together with a negative control (BSA), showed a much lower response than Con A [94].

The characteristics of AuNP-based glycan biosensors are summarized in Table 1.

### 2.2. Quantum Dots (QDs)

Quantum dots (QDs) are very small (2 to 10 nm in diameter), semiconductor particles, used for imaging, sensors and cell targeting [56,95]. The optical and chemical properties of QDs, such as fine fluorescence wavelength, symmetrical fluorescence emission peak, quantum yield, brightness and photostability depend on the size of the NPs [96]. Unfortunately, QDs are predominantly prepared in a non-polar organic solvent, resulting in them being hydrophobic with a tendency to irreversibly aggregate with a poor colloidal stability in aqueous media. Consequently, biologic utilization is limited [97] but, in recent years, this problem was resolved and QDs were widely used as fluorophores for the visualization of a number of biologic and chemical processes [72,98], including biosensing applications [99], imaging of bacteria and live cells [100,101], targeting cancer cells [102] or for designing gene delivery systems [103]. Glycan-coated QDs were recognized as multivalent tools to display protein–carbohydrate interactions associated with inter- and intracellular recognition processes [104].

A simple and rapid test for detection of the cholera toxin from *Vibrio cholerae* was developed by Lee’s group [105]. A complex of thiol-modified β-Gal derivatives immobilized on 10 nm AuNPs and amino-terminated QDs was produced, because of the strong hydrogen bonds between the QDs amines and galactose hydroxyl groups. In the presence of the analyte (toxin), β-Gal-AuNPs recognized the cholera toxin, inhibiting the formation of a QD complex and, consequently, recovered fluorescence [105].

Zhang’s team [106] examined commercially available CdSe/ZnS-COOH QDs with covalently attached glucosamine (λ_em_ = 525 nm) and galactosamine (λ_em_ = 605 nm) moieties, respectively. Individual complexes were applied to simultaneous dual-color quantitative analysis of Con A specific to Glc and PNA interacting with Gal. After interaction of the conjugates with the analytes, aggregation was observed and, consequently, a decrease in the fluorescence of QDs in the supernatant was determined. The authors declared LOD of 0.3 nM (Con A) and 0.18 nM (PNA) [106].

A quadruple-channel optical nanosensor with green (G), yellow (Y) and red (R) emissions was created by Wang and co-workers [107]. A nanoprobe consisting of CdSe/ZnS QDs and saccharides was prepared (Man–G–QDs, GlcNAc–G–QDs, Gal–Y–QDs, GlcNAc–Y–QDs, Man–R–QDs and Gal–R–QDs) and used in the analysis of five lectins (Con A, WGA, PNA, RCA and PSA), all of them with different properties (i.e., molecular weight/mass, number of subunits and binding specificity) with the same concentration of 1 M. Lectins could be detected at the level of an individual lectin due to their different binding constants with saccharide units using distinct quadruple-channel signals. QDs were used for FRET detection in three channels and the fourth channel was used for Rayleigh resonance-scattering optical detection. The biosensor was prepared by the immobilization of thiolated monosaccharides to QDs via a metal–sulfur bond. The signals obtained were displayed as a heat map with the application of a linear discrimination analysis to multiplexed quantitative analysis. The size of the particles increased from 24 nm to 611 nm, when detecting Con A and the biosensor offered LOD for five lectins within the range of 4.6–9.7 nM, depending on the lectin analyzed [107].

Another multiplexed analysis of lectins implementing QDs was based on the integration of 2D nanomaterial MoS_2_ (50–200 nm × 1.2 nm) [108]. Three different CdSe/ZnS QDs were modified by thiolated monosaccharides as three independent fluorescent probes with emissions at 520 nm, 580 nm and 660 nm with their sizes ranging from 8 to 18 nm. Their emission was quenched by incubation with MoS_2_ nanoplatelets modified by phenylboronic acid. Upon interaction with five lectins or 2 bacterial strains with fluorescent probes, their distance from MoS_2_ nanoplatelets increased, resulting in a recovered fluorescent signal (Figure 9). The results were again represented as a heat map with a linear discrimination analysis applied to data evaluation. The biosensor could detect lectins with LOD down to low nM range and bacteria down to 87 CFU·mL^−1^ (*E. coli*) or 66 CFU·mL^−1^ (*Enterococcus faecium*) [108]. Zhang’s group [109] used a similar approach by the quenching of a fluorescent signal generated by QDs using AuNPs for detection of the Con A lectin. In this study, glycans were used as a recognition element for detection of the glycan-binding proteins which are present on many walls of microbes. The core of QDs with emission at 545 nm was modified by mannose to produce a Man-QD probe for detection of a lectin with LOD of 3 nM [109].

CdSe/ZnS QDs were also used in the detection of fluorescently labeled lectins via FRET with two methods of glycan attachment to QDs shown, but with no analytical parameters provided [110].

The characteristics of QDs-based glycan biosensors are summarized in Table 2.

### 2.3. Magnetic Nanoparticles (MNPs)

The magnetic properties of biologic analytes and samples are usually reduced to a minimum. This leads to high selectivity and a signal-to-noise ratio of a magnetic separation even in complex-sample mixtures. Additionally, it is possible to use MNPs for analyte preconcentration introduced prior to analysis itself with this step boosting the sensitivity of analyte detection [65].

Rouhanifard and colleagues [111] showed an application of glycan-MNPs for cell enrichment and presented it as an effective, cheap and more suitable approach than conventional antibody-based methods for the determination of dendritic cells (DCs, Figure 10). The role of such cells within an immune system is for antigen-presenting, with responsibility for communication between innate and adaptive immunity and for either exciting or prohibiting immune reactions [111,112].

A polyacrylic acid linker was used for modification of MNPs (Fe_3_O_4_ core; d = 10 nm) to create colloidal nanocrystal clusters (200–300 nm) [111]. An array of techniques has been used for the characterization of prepared MNPs. Specificity, biocompatibility and the feasibility of using Lewis x antigen (Le^x^)-linker-MNPs for capturing DC-SIGN (Dendritic Cell-Specific Intercellular adhesion molecule-3-Grabbing Non-integrin, it is a C-type lectin receptor) have been investigated. DC-SIGN belongs to the class of C-type lectin receptors, which is located on both (immature and mature) DCs derived from monocytes, lymph nodes, spleen and tonsils. This makes it a very specific marker for targeting the cell populations described above. DC-SIGN specifically recognizes glycosylated mannose- or fucose-rich antigens, such as those found in the bloodstream, including Le^x^. Glycan-functionalized MNPs did not interfere with the cell activity. Finally, the authors examined an application of Le^x^-functionalized MNPs to the enrichment of differentiated and immature DCs from a mixed population of human cells. MNPs with the Le^x^ epitopes were not only biocompatible, but this study was the first to display the usefulness of glycans as ligands for DC capture and isolation [111].

Tuberculosis (TB) is one of the top ten causes of death worldwide. To date, sputum smear microscopy is used as the main method for detecting pulmonary TB. It is a very simple, fast and cheap method, but with a limited sensitivity. Bhusal’s team [113] decided to focus on the development of a nanoparticle-based colorimetric biosensing assay for cheap and rapid detection of very low concentrations of acid-fast bacilli (AFB) in sputum samples. Bhusal’s team created MNPs covered with immobilized glycan, which can bind to the bacterial cell wall by glycan-binding proteins. The core of MNPs was composed of iron oxide (III), which was wrapped in a chitosan layer. The average size of these superparamagnetic particles was (99 ± 58) nm. Glycans on the surface permit the inexpensive and simple capture of *Mycobacterium tuberculosis* cell without using antibodies or aptamers. By using modified MNPs, the authors were able to make a complex MNPs–AFB, which could be isolated and concentrated from the rest of the sample by a magnetic field. The analysis was completed within 20 min, with an estimated cost per analysis of $0.10 and with LOD of 100 CFU·mL^−1^ for bacteria [113].

The characteristics of MNP-based glycan biosensors are summarized in Table 3.

### 2.4. Carbon Nanoparticles

A significant drawback considerably limiting the application of carbon nanotubes (CNTs) in life sciences is their low solubility in aqueous solutions and the formation of stacked and interconnected individual nanotubes forming rope-like 3D aggregates [54]. There are various methods (covalent, non-covalent or hybrid) for modifying CNTs to render them dispersible in aqueous solutions. Covalent modification techniques are more frequently applied, since such modifications are stable, but these modifications can affect the electronic properties of CNTs. Non-covalent modification protocols represent the only option for applications requiring CNTs without altered electronic properties. Hence, a particular modification of CNTs has to be subject to informed selection, taking the final application of CNTs into account.

The influence of carbohydrate density and spatial configuration on the binding of Con A was investigated using 3 different nanoscaffolds deployed for carbohydrate immobilization (Figure 11) [115]. All three nanoscaffolds (Borromean rings, dodecaamine cages and fullerenes) could accommodate 12 carbohydrate units with different spatial configuration and density. Quartz crystal microbalance (QCM) was used to investigate the binding kinetics of Con A towards such immobilized carbohydrates with significant differences in the association phase, dissociation phase and affinity constants (Figure 12). The greatest affinity was observed for molecular Borromean rings (147 nM), followed by fullerenes (1.02 M) and dodecaamine cages (1.21 M). The differences in the binding were explained by the more optimal local densities of carbohydrates on the molecular Borromean rings, in contrast with the higher spatial dispersion of carbohydrates on the dodecaamine cages and the smaller, more constrained fullerenes. These results reveal the importance of the spatial configuration and density of immobilized carbohydrates for recognition events [115].

Carbon dots (CDs) are relatively new nanomaterials that have attracted interest due to their unique optical characteristics, which are similar or sometimes superior to semiconductor QDs. Their photoluminescence properties, chemical inertness, excellent water solubility, low manufacturing costs and general minimal toxicity suggest they possess extensive potential applications [72]. Carbohydrate-functionalized CDs were used by Swift et al. [116] to probe interactions with biologic systems, with the conclusion that the choice of carbohydrate used in the modification of CDs significantly affected their electronic properties.

Glycan-coated CDs were produced by Hill et al. [117]. Hill’s group prepared and characterized lactose-coated CDs with an extremely low confirmed toxicity of novel Lac-CDs and, using confocal microscopy, they were able to visualize the Lac-CDs’ interaction with HeLa and MDA cells. The cells were visualized using excitation at 405 nm and the results proved that the Lac-CDs were internalized by both cell lines [117]. Swift et al. prepared several glycan nanoprobes (glucose, mannose, galactose, maltose and lactose) based on CDs. Their work contains an extensive characterization from the materials’ perspective. Following the formation of various CDs using different carbohydrates, they were able to demonstrate how selection of the carbohydrate functionalization altered the electronic structure of the interfacial layer of CDs [116].

Martin’s group [118] focused on the synthesis of mannose-coated dynamic and static micelles from diacetylene-derived mannopyranosyl glycolipids, which were self-assembled on SWCNTs (see Figure 4). First, the availability of the sugar epitopes was confirmed by using mannose specific lectin (Con A). The binding of Con A to such micelles was performed using an inhibition assay. Mannose-containing micelles were investigated as inhibitors of an interaction between Con A and HRP (heavily mannosylated) in an enzyme-linked lectin assay. The results showed the following inhibition constant (IC_50_): 1170 μM for α-D-mannopyranoside; 113 μM for dynamic micelle (non-cross-linked); 46 μM for static micelle (cross-linked) and 0.5 μM for glyconanoring around SWCNTs [118]. The same group later investigated the effect of the length of polyethylene glycol and the redox state of sulfur within diacetylene-based mannopyranosyl glycolipids on the affinity to Con A [119].

Fullerene with immobilized 12 mannose units was used in the interaction with lectins, but only K_D_ values were obtained using a quartz crystal microbalance device [120].

In our work, we developed a label-free electrochemical approach for the detection of an antibody against an aberrant form of a glycan—Tn antigen [121]. In this case, the Tn antigen was covalently attached to a pre-immobilized layer of BSA on the surface of an electrochemically activated screen-printed graphene electrode. The electrochemical oxidation of the graphene interface had two distinct roles: a) to form –COOH groups applied to covalent immobilization of BSA via amine coupling; and b) to render the surface more hydrophilic and thus more accessible for the aqueous assay buffer and proteins examined (antibody and lectin). The study showed that the lectin could be detected down to 1 aM, while the antibody against the Tn antigen down to 10 aM [121]. The main aim of the study was to develop an ultrasensitive biosensor device for the detection of antibodies against aberrant glycans, which can be effective used as biomarkers in the diagnosis of several types of cancers [122].

Another graphene-based device was constructed by immobilizing carbohydrates onto a graphene surface via π–π stacking of the anthraquinone-carbohydrate conjugate [123]. Anthraquinone was used as a redox probe with measurement of an electrochemical signal generated by this interfacial redox probe, while a soluble redox probe was also present in the system. The electrochemical detection was evaluated as a ratiometric signal, i.e., as a ratio of the peak current of a soluble redox probe divided by the peak current of anthraquinone (a surface-confined redox probe). An affinity interaction (i.e., Con A or various cells) only changed the peak current of the soluble redox probe with Con A detected down to nM level [123].

The characteristics of carbon nanoparticle-based glycan biosensors are summarized in Table 4.

### 2.5. Other Nanoparticles

Polystyrene particles with size exceeding 100 nm (i.e., ~650 nm) have been used to form a 2D array on the surface of glass [125]. Such particles were able to detect shrinkage of a carbohydrate-containing hydrogel upon interaction with lectins down to a concentration of 75 nM for ricin, 230 nM for jacalin and 38 nM for Con A, when exposed to lactose-, galactose- and mannose-containing hydrogels. The shrinkage of the hydrogel was observed as a change in the interparticle spacing over distances up to 50 nm (Figure 13) [125].

A supramolecular self-assembly of the conjugate, consisting of an oligo ethylene glycol linker, a binder (i.e., a pyrene moiety) to 2D MoS_2_ nanomaterial and a glycoligand on 2D MoS_2_ nanomaterial, was used for detection of a lectin using electrochemistry [126]. A decrease in the differential pulse voltammetric peak current was observed upon binding of a lectin (LOD of 373 nM) and human hepatoma cancer cells (LOD of 840 cells·mL^−1^) [126].

2D MoS_2_ nanomaterial was also used in the construction of a fluorimetric biosensor device [127]. In that case, a conjugate consisting of an oligo ethylene glycol linker, a glycoligand and a fluorescent probe (having an affinity towards MoS_2_ nanomaterial) was incubated with MoS_2_ nanosheets, which quenched the fluorescent signal. Two-dimensional glycosheets were prepared by incubation of two different glycoconjugates, i.e., glycoconjugate 1 containing α-2,6-sialyllactose and fluorophore 1 and glycoconjugate 2 containing α-2,3-sialyllactose and a fluorophore 2. Upon binding with influenza viruses such as H1N1 (α-2,6-SA binding), glycoconjugate 1 was released from the 2D glycosheets restoring fluorescence due to the release of fluorophore 1. Upon binding with the H10N7 influenza strain the signal was generated by fluorophore 2 and when the influenza strain H7N9 recognizing both glycoconjugates was incubated with 2D glycosheets, the fluorescence of both fluorophores was restored (Figure 14**)**. The LOD for each particular influenza strain was 8 (H1N1), 4 (H10N8) and 14 (H7N9) hemagglutinin units per mL. The response was obtained within 5 min [127].

### 2.6. Hybrid Nanoparticles

Cancer cells, like normal ones, are covered with many specific molecular structures consisting of various glycoproteins which can be recognized by specific carbohydrates. Guan et al. [128] formed glycosylated liposomes with CDs (~5.5 nm) loaded into the interior of liposomes (~80 nm), while mannose units were attached to the surface of the liposomes for targeted recognition of HepG2 cells (liver cancer cells, Figure 15). The HepG2 cells were recognized via carbohydrate–glycoprotein interactions. The main reason for encapsulation of the CDs into liposomes was their increased stability and fluorescence intensity [128].

### 2.7. Other Nanoscale Approaches

The use of immobilized glycans as bioreceptive molecules for the detection of influenza viruses is based on the natural interaction between the hemagglutinins of influenza viruses and the sialic acid receptors on the surface of the host cells [129]. An organic electrochemical transistor consisting of a conductive polymer channel with immobilized trisaccharide α-2,6-sialylactose was used in the detection of influenza virus A [130]. An unmodified glycan was covalently grafted onto the conductive polymer by an oxime grafting protocol (Figure 16). Upon binding with negatively charged influenza virus A, the polymer was negatively doped, resulting in changes in the drain current of the device. LOD for virus detection was 0.025 HAU, which was two orders of magnitude lower than with conventional rapid human influenza virus test kits [130]. A similar two-channel device based on a semiconductor interface for immobilization of α-2,6-sialylactose and α-2,3-sialylactose was used for the detection of human (H1N1) and avian (H2N1) influenza viruses [131]. The portable device connected to a mobile phone via Bluetooth could measure the interaction in a label-free format with the integration seen in real time (Figure 17). The device could detect influenza viruses at a concentration much lower than by using a device based on immunochromatography [131]. Another device for the detection of influenza relies on the compression of a polymer layer resulting in the induction of stress in the artificial layer causing a resonance frequency shift (Figure 18) [132].

The characteristics of nanoscale-based glycan biosensing platforms are summarized in Table 5.

## 3. Proteins as Nanoscaffolds

The multivalent nature of carbohydrate–lectin interactions is crucial because saccharide units immobilized on a biosensor surface are required to have an appropriate lateral density and structural formation to mimic the cell surface’s glycocalyx. It is easier to attach large polysaccharides onto different interfaces than to attach small hydrophilic glycan probes. The large ones are immobilized on interfaces either via hydrophobic interactions or van der Waals forces. On the other hand, small saccharide molecules can be attached via functional groups (–SH, –NH_2_) or in the form of a conjugate with various scaffolds, such as dendritic polymers, DNA, lipids and proteins [133].

Since, in conventional assay formats like ELISA, proteins (bovine or human serum proteins in particular) are applied to decrease non-specific binding, when using complex samples such as human serum, such proteins can also act as natural scaffolds for the attachment of small glycans. Kveton et al. [134] compared three techniques for glycan immobilization on the surfaces of planar polycrystalline gold electrodes. Two methods were based on the binding of the Tn antigen to form 2D biosensors and the third used a layer of a globular protein (HSA), forming a 3D configuration on the surface of the biosensor. Electrochemistry was applied as a transducing platform to investigate the binding of an antibody against such an aberrant glycan. A 3D biosensor based on HSA afforded a much better analytical performance than the 2D configuration in terms of sensitivity, linear range and LOD (1.4 aM vs. 270 aM). The study also showed that the 3D structure-based sensor was not only more sensitive than the other two 2D biosensor platforms, but that the Tn antigen on the 3D biosensor surface was more accessible for analyte binding with better kinetics of binding [134].

The correct localization of glycans and the controlled immobilization on protein probes play key roles in the construction of glycan biosensors. Tao et al. [133] produced a series of BSA-sugar conjugates with a variable number of mannose moieties. The BSA conjugates were attached directly onto a gold substrate without any interfacial activation, taking into account the strong adsorption of proteins on gold. Three derivatives of BSA were prepared, differing in the number of mannose units attached to BSA (8, 11 or 13 mannose molecules) in a site-specific manner using a squaric acid conjugation between lysine molecules of BSA and –NH_2_ group of hexylamine-modified mannose. With increasing the number of mannose units on BSA, a lower amount of the conjugate was bound to the gold surface, which was explained by the increasing hydrophilicity of the conjugate with an increased number of mannose units. SPR measurements revealed that the conjugate BSA-Man with 11 mannose moieties had the highest affinity towards Con A with LOD of 1.8 nM. The regeneration and specificity of the obtained glycan biosensors were also investigated [133]. When the BSA was immobilized on the gold surface first and the BSA layer was then modified with mannose units via a squaric acid conjugation, the SPR biosensor could detect Con A down to 1.9 nM [135]. The same approach with BSA modification with mannose units using a squaric acid conjugation was deployed in fluorescent microarray experiments when, besides the mannose units, a fluorescent probe was also attached to BSA [136].

BSA-based neoglycoproteins were prepared by the conjugation of NH_2_-modified glycans to BSA using squaric acid [137]. The neoglycoproteins were then immobilized in wells on a microtiter plate. Complex forms of the glycans were then prepared by elongation of the glycan part of the neoglycoprotein using recombinant glycosyltransferases, resulting in an array comprising 40 different glycan epitopes based on *N*-acetyllactosamine. This array was then probed using *Clostridium difficile* (causing gastrointestinal disorders) enterotoxin A and B, which were detectable down to nM concentration [137].

The characteristics of protein nanoscaffold-based glycan biosensing platforms are summarized in Table 6.

## 4. Conclusions

Most approaches described here are based on the conjugation of a simple monosaccharide with various types of nanomaterials with the subsequent detection of lectins, cells or influenza viruses to prove viability of the proof of concept. In order to render glyconanobiosensors suitable for the analysis of a diverse range of glycan-binding proteins, either in isolation or present on the surface of various types of cells or viruses, there is a requirement to immobilize complex glycans, which occur naturally. At the same time, it is very important to use such nanobiosensors for the detection of analytes, which can become disease biomarkers such as, for example, antibodies against aberrant glycans. The other issue to be addressed is to design glyconanobiosensors affording a limit of detection well below nM level, since disease biomarkers can be present in body fluids at an early stage in concentrations at, i.e., pM levels. In order to make glyconanobiosensors a viable alternative to the frequently applied glycan arrays, it is of the utmost importance to design glyconanobiosensors so as to allow at least a moderate level of assay throughput.

## Figures and Tables

**Figure 1 nanomaterials-10-01406-f001:**
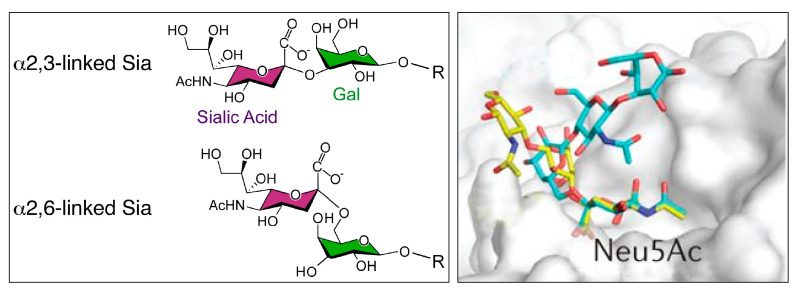
(**Left**): Structure of *N*-acetylneuraminic acid (Neu5Ac, sialic acid, SA), linked to galactose either via a α-2,3- or via a α-2,6-linkage. Figure taken from [18], which is an open access article distributed under Creative Commons attribution license. (**Right**): Superposition of an avian influenza virus hemagglutinin in a complex with α-2,3-sialyllactosamine (yellow) (PDB accession 2WR2) and the human receptor α-2,6-sialyllactosamine (cyan) (PDB accession 2WR7). The avian receptor generally has a linear conformation, whereas the human receptor is more flexible and has an umbrella-like topology. Reproduced with permission from [14]. Copyright Nature, 2014.

**Figure 2 nanomaterials-10-01406-f002:**
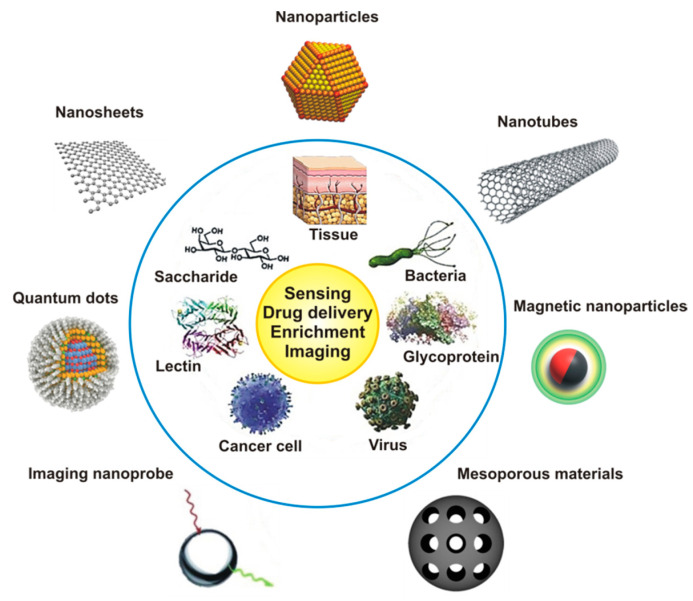
Overview of different nanomaterials (outside the blue circle) used in combination with biomaterials/biomolecules (inside the blue circle) for various applications (in yellow circle). Reproduced with permission from [37]. Copyright Elsevier, 2014. Figure redrawn in our review paper [9], which is an open access article distributed under the terms of the Creative Commons CC BY license.

**Figure 3 nanomaterials-10-01406-f003:**
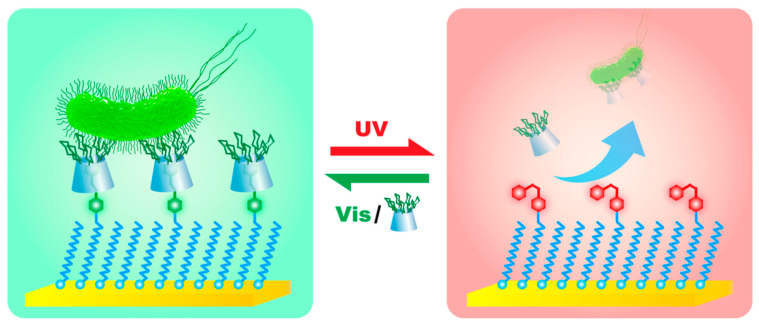
To switch the bacteria-capture state (green part, trans- form of Azo) to the bacteria-release state (red part, cis-form of Azo), the gold surface covered by azo derivative hosting β-cyclodextrin with covalently attached mannose units, was irradiated with 36 nm UV light for 30 min. To switch the bacteria-release state back to bacteria-capture state, the interface was irradiated with 450-nm visible light for 30 min and then immersed it into a fresh cyclodextrin-mannose solution. Reproduced with permission from [47]. Copyright American Chemical Society, 2017.

**Figure 4 nanomaterials-10-01406-f004:**
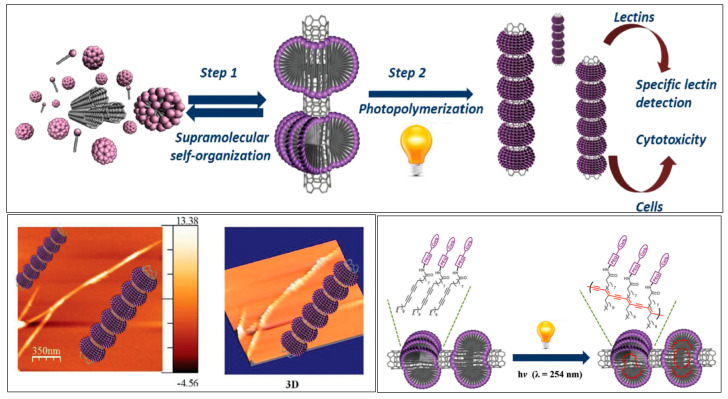
Synthesis and applications of hierarchically self-assembled and photopolymerized neoglycolipids onto SWCNT sidewalls (**upper**). SEM and TEM images of SWCNTs modified by glycolipids forming nanorings (**lower left**) and polymerization of the glyconanoring-coated carbon nanotubes (**lower right**). Reproduced with permission from [54]. Copyright Royal Society of Chemistry, 2015.

**Figure 5 nanomaterials-10-01406-f005:**
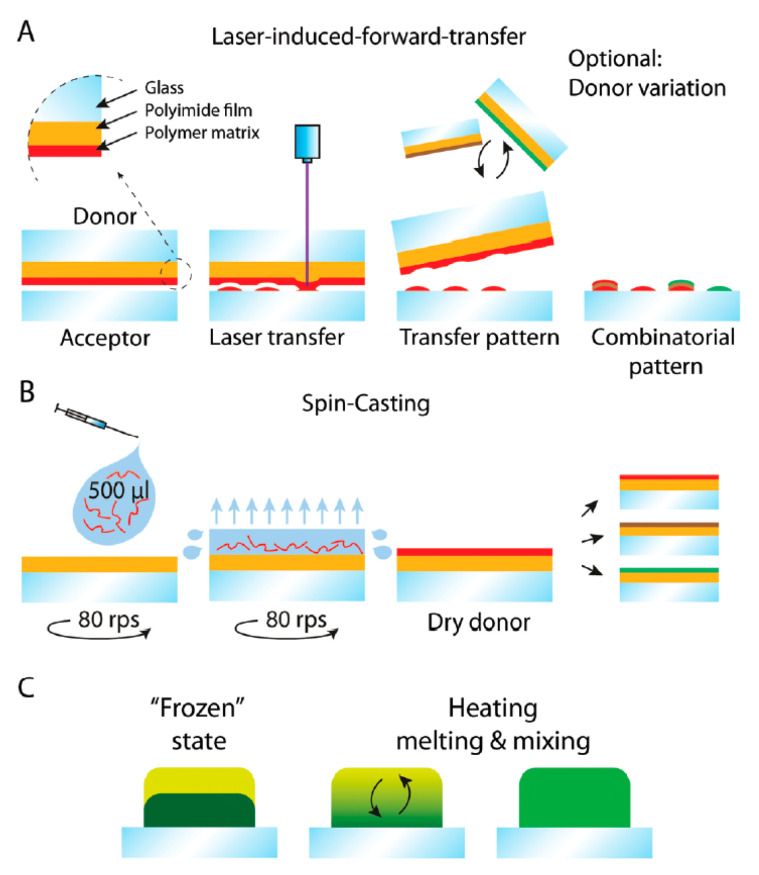
General procedure. (**A**) Microarray generation with laser induced forward transfer; (**B**) Spin-coating of donor slides (glass slide ≈1 mm, covered with self-adhesive polyimide foil, ≈75 um) with different polymer mixtures (e.g., polystyrene mixed with amino acid building block or fluorescent dye); (**C**) Melting and mixing of transferred polymer spots (not to scale). Reproduced with permission from [55].

**Figure 6 nanomaterials-10-01406-f006:**
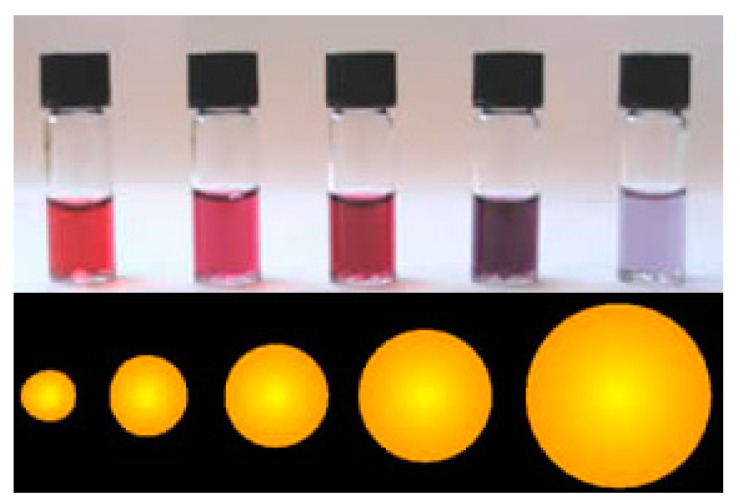
Color differences caused by variations in size of AuNPs. Figure taken from Wikipedia on 6 April 2020 [79].

**Figure 7 nanomaterials-10-01406-f007:**
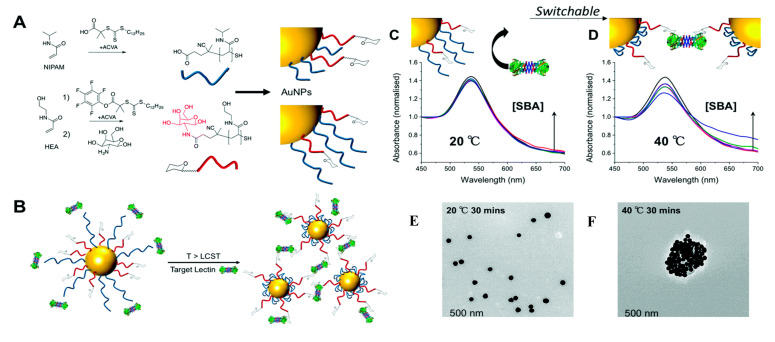
(**A**) Synthesis of polymers by RAFT (reversible activation fragmentation transfer); (**B**) concept of using responsive polymers to gate access to nanoparticles. Below the critical temperature longer polymer chain due to steric hindrance prevents lectin binding to glycans, but above the critical temperature, the polymer collapse to expose glycans enabling binding and aggregation of the particles. UV-Vis traces of different nanoparticle formulations in presence of serial dilution of lectin (1–10 μg·mL^−1^) after 30 min incubation at 20 °C (**C**) or 40 °C; (**D**). An increase in Abs_700_ and decrease in Abs_540_ is indicative of binding. All curves were normalized to Abs_450_ = 1. TEM images of these particles after addition of lectin (**E**) at 20 °C for 30 min; (**F**) at 40 °C following 30 min incubation. Reproduced with permission from [88]. Copyright Royal Society of Chemistry, 2017.

**Figure 8 nanomaterials-10-01406-f008:**
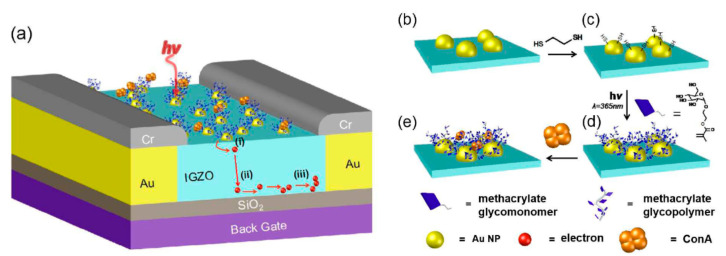
(**a**) A plasmonic field effect transistor (FET) for sensing lectins is coated with a layer of AuNP-glycopolymer conjugates. Increases in plasmon absorption upon the binding of lectins to the glycopolymers increases the electrical current (dimensions are not proportional): (i) Hot electrons transfer from AuNPs to the InGaZnO (IGZO) layer by overcoming the Schottky barrier; (ii) The applied gate voltage assists the migration of hot electrons into the current channel of the plasmon FET; (iii) The number of charge carriers increases in the IGZO layer, thereby enhancing the detected signal. Sensor fabrication: (**b**) AuNPs are assembled onto the plasmon FET; (**c**) 1,2-ethanedithiol is adsorbed onto the Au NPs; (**d**) glucosyloxyethyl methacrylate is incubated onto the Au NP under UV light (λ = 365 nm) in the presence of a photoinitiator for 2 h to form the glycopolymer via a thiol-acrylate photopolymerization; (**e**) binding of the lectin Con A to the glycopolymers on the AuNP surfaces. Symbols for each of the elements in the figure are provided at the bottom. Reproduced with permission from [92]. Copyright the Royal Society of Chemistry, 2016.

**Figure 9 nanomaterials-10-01406-f009:**
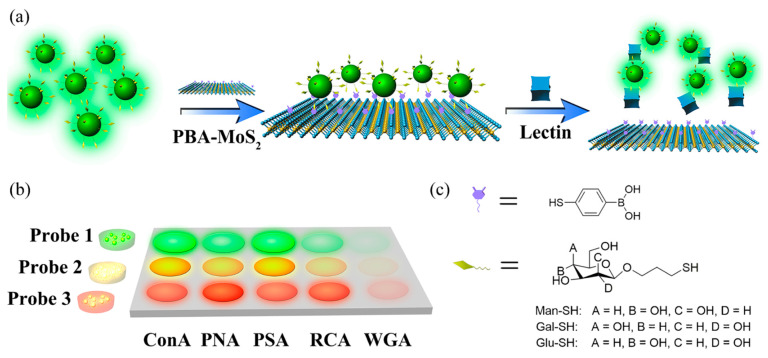
(**a**) Schematic illustration of the rational design for lectin detection; (**b**) construction of a fluorescent array by employing Man-QD_520_ (Probe 1), Glu-QD_580_ (Probe 2) and Gal-QD_660_ (Probe 3) as three signal probes; (**c**) molecular structures of saccharide derivatives for QDs modification. Reproduced with permission from [108]. Copyright Springer, 2018.

**Figure 10 nanomaterials-10-01406-f010:**
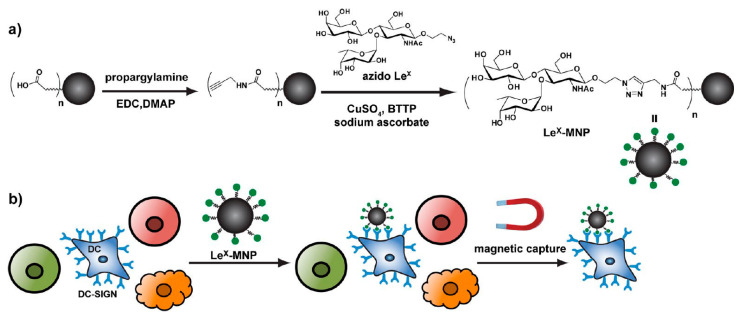
(**a**) synthesis of Le^x^-coated MNPs and (**b**) selective capturing of DCs from a cell population by functionalized MNPs. Reproduced with permission from [111]. Copyright American Chemical Society, 2012.

**Figure 11 nanomaterials-10-01406-f011:**
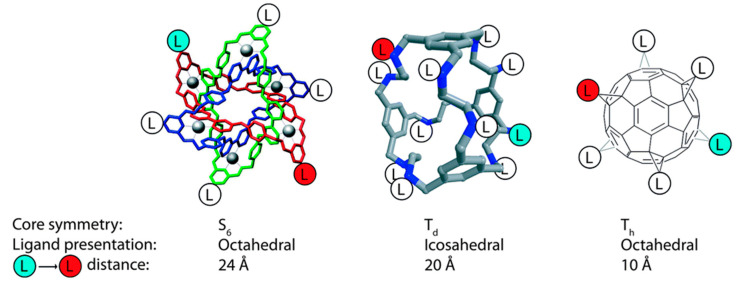
Symmetry and size of nanoscaffolds (from left to right): molecular Borromean rings, dodecaamine cages and fullerenes applied for glycan immobilization with subsequent examination of kinetic and affinity parameters for binding to Con A. Reproduced with permission from [115]. Copyright the Royal Society of Chemistry, 2016.

**Figure 12 nanomaterials-10-01406-f012:**
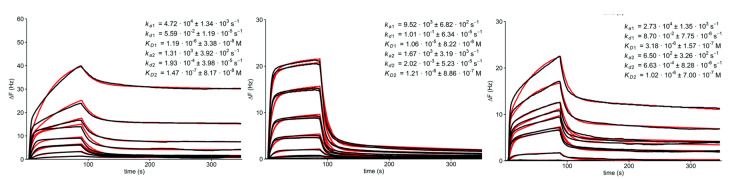
QCM sensorgrams (black curves) and fitted datasets (red curves) for (from left to right): molecular Borromean rings, dodecaamine cages and fullerenes together with estimated binding data. Reproduced with permission from [115].

**Figure 13 nanomaterials-10-01406-f013:**
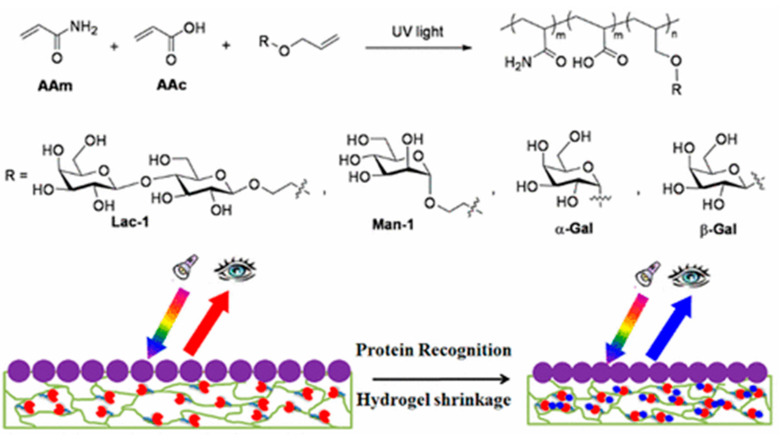
Shrinkage of the carbohydrate-containing hydrogel upon binding of a lectin detected as a change of interparticle distance diffracting visible light. Reproduced with permission from [125]. Copyright American Chemical Society, 2017.

**Figure 14 nanomaterials-10-01406-f014:**
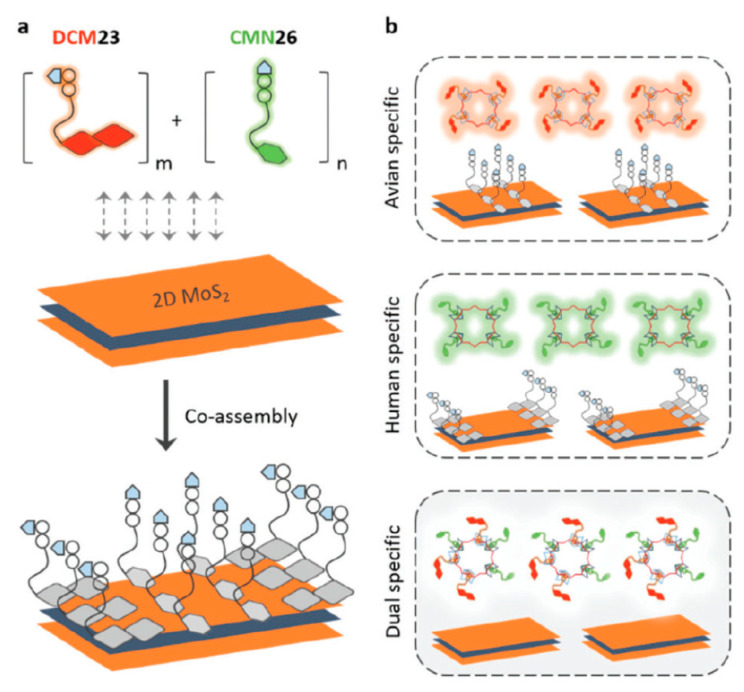
Schematic illustration of (**a**) the co-assembly of sialyl–glycan–fluorophore conjugates with different emission colors to 2D MoS_2_, producing the 2D glycosheet and (**b**) the use of the 2D glycosheet for identification of the single or dual receptor specificity in influenza viruses. Reproduced with permission from [127]. Copyright the Royal Society of Chemistry, 2017.

**Figure 15 nanomaterials-10-01406-f015:**
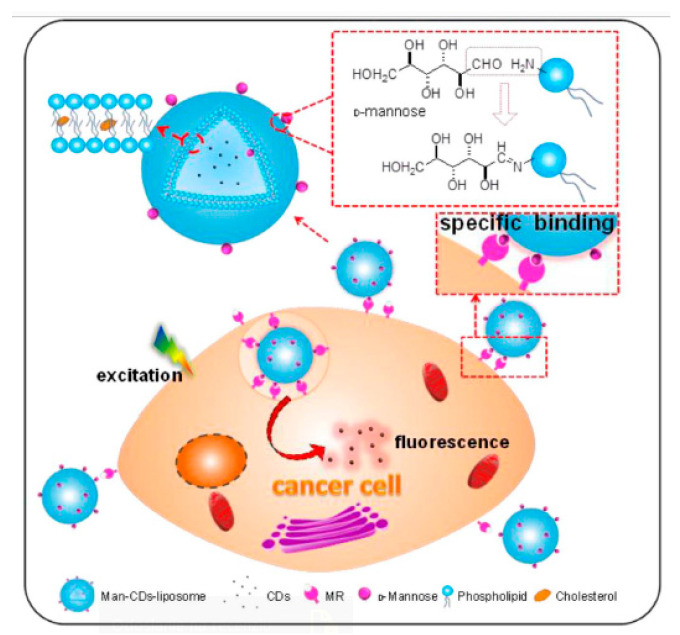
Schematic illustration depicts the structure of Man-CDs-liposome. The process for Man-liposomes loading is displayed. The aldehyde groups of D-mannose can react with the amino on the hydrophilic head of liposomes to form imine by aldehyde-amide condensation reaction. Man-CDs-liposome can target recognition of HepG2 cells. Reproduced with permission from [128]. Copyright Elsevier, 2018.

**Figure 16 nanomaterials-10-01406-f016:**
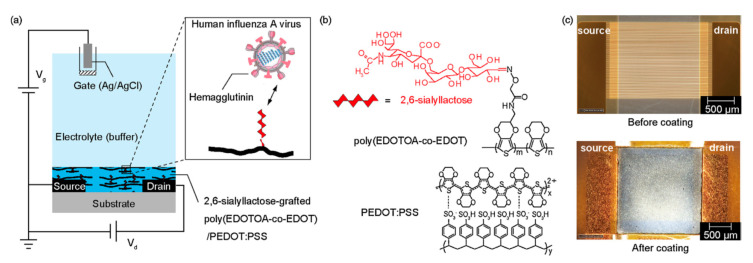
An organic electrochemical transistor device for specific human influenza A virus sensing. (**a**) Schematic cross section of the organic electrochemical transistor device with a thin layer of 2,6-sialyllactose-grafted polymer in an electrolyte; (**b**) chemical structures of 2,6-sialyllactose-grafted polymer; (**c**) optical micrographs showing top view of the interdigitating drain–source microelectrodes before and after the coating of a polymer thin film. Scale bar: 500 um. Reproduced with permission from [130]. Copyright Elsevier, 2018.

**Figure 17 nanomaterials-10-01406-f017:**
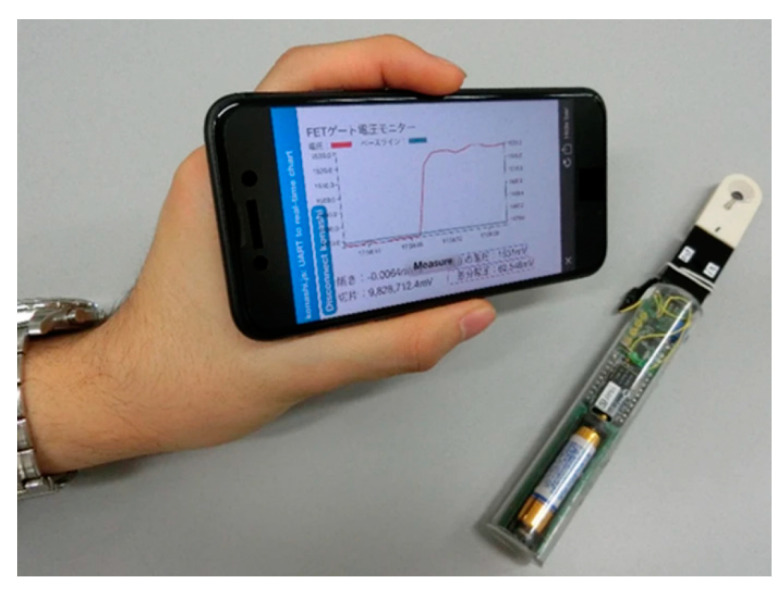
Portable biosensing system using a Bluetooth connection between a smartphone and the field-effect transistor-based biosensor. Reproduced with permission from [131].

**Figure 18 nanomaterials-10-01406-f018:**
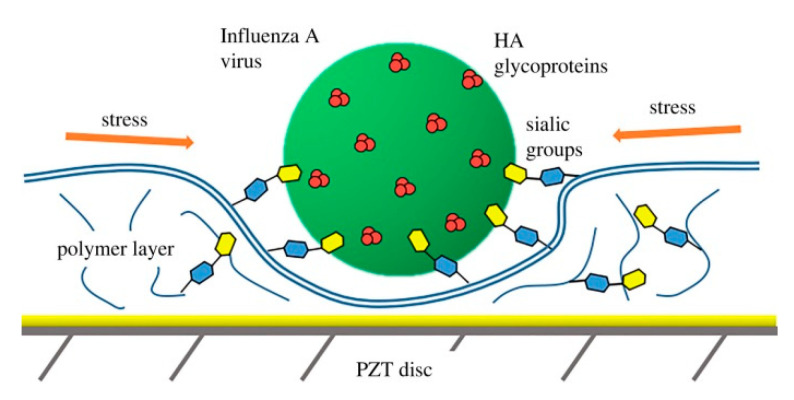
Rise of lateral stress inside the polymer receptor layer with sialyloligosaccharide groups after binding virus particle. Reproduced with permission from [132]. Copyright the Royal Society, 2019.

**Table 1 nanomaterials-10-01406-t001:** AuNP-based glycan nanobiosensors and their characteristics.

Carbohydrate	Immobilization	Interface	Transducer	Analyte	LOD	Ref.
SA residues	thiol–Au–S bond	AuNPs	optical (96-well plate)	influenza viral strains	8 HA titer	[69]
lactose glycoconjugate	thiol 4-mercaptobenzoic acid		surface- enhanced Raman scattering	galectin 9	1.2 × 10^−9^ M	[78]
fetuin with SA	fetuin micelle		carbon SPE	influenza strain H9 N2	8 HAU titer	[80]
α-D-mannose and β-D-galactose dendrons	thiol	spherical AuNPs	optical (96-well plate)	*E. coli* strains ORN 178	200 μg·mL^−1^	[82]
		rod-shaped AuNPs			20 μg·mL^−1^	
galactose and mannose	click chemistry (PEG)	spherical AuNPs			17 ± 2 μg·mL^−1^	[83]
		rod-shaped AuNPs			14 ± 2 μg·mL^−1^	
		star-like AuNPs			0.03 ± 0.01 μg·mL^−1^	
mannose	UV-irradiation	InGaZnO gate	FET	Con A	10^−10^ M	[92]
galactosamine and mannosamine	terminal thiol from RAFT agent	AuNPs	optical (96-well plate)	carbohydrate-binding proteins	down to nM	[85]
ovalbumin with mannose, glucose and Galβ(1→4)GlcNAc	cysteines residues		MALDI	Con A	3.9 nM	[86]
				BanLec	7.8 nM	
				ricin B	31.3 nM	
maltose	NaOH, 50 °C		optical (96-well plate)	Con A	23 pM	[87]
battery of saccharides	terminal thiol from RAFT agent			battery of lectin and Ca^2+^	μg·mL^−1^	[88,89]
4-aminophenyl α-D-mannopyranoside and 4-aminophenyl β-D-galactopyranoside	click chemistry (PEG)	Au nanorods	optical (near-infrared absorption and scattering in surface plasmon resonance)	*E. coli*	down to μM	[91]
mannose	thiols	Ag-coated Au nanorods	optical (SPR)	Con A	down to nM	[77]
					2 nM	[94]

**Table 2 nanomaterials-10-01406-t002:** QDs-based glycan nanobiosensors and their characteristics.

Carbohydrate	Immobilization	Interface	Transducer	Analyte	LOD	Ref.
β-galactose derivatives	thiols	AuNPs and amino-terminated QDs	optical (fluorescence resonance energy transfer)	cholera toxin from *V. cholerae*	280 pM	[105]
glucosamine	amide coupling reaction	CdSe/ZnS QDs	optical (dual-color quantitative analysis)	Con A	0.3 nM	[106]
galactosamine				PNA	0.18 nM	
mannose, galactose, *N*-acetylglucosamine	thiols (metal–sulfur bond)		optical (fluorescence resonance energy transfer)	Con A	4.6 nM	[107]
				WGA	9.7 nM	
				PNA	8.9 nM	
				RCA_120_	5.7 nM	
				PSA	4.6 nM	
		PBA-MoS_2_ + CdSe/ZnS QDs	optical	Con A	3.7	[108]
				PSA	8.3	
				PNA	4.2	
				RCA_120_	3.9	
				*E. coli*	87 CFU·mL^−1^	
				*Enterococcus faecium*	66 CFU·mL^−1^	
*N*-linked glycan terminated with sialic acid	covalent bioconjugation through oxime ligation	CdSe/ZnS QDs	optical (fluorescence resonance energy transfer—FRET)	SNA	undefined	[110]
	His-tag self-assembly					
α-mannose	thiols -	CdSe/ZnS QDs	optical (FRET)	Con A	3 nM	[109]

**Table 3 nanomaterials-10-01406-t003:** MNPs-based glycan nanobiosensors and their characteristics.

Carbohydrate	Immobilization	Interface	Transducer	Analyte	LOD	Ref.
Lewis x	polyacrylic acid + CuAAC chemistry	MNPs (Fe_3_O_4_ core)	magnetic field	dendritic cells	undefined	[111]
*N*-glycans	boronate affinity controllable oriented surface imprinting	MNPs	systematic evolution of ligands by exponential enrichment	aptamers	<1 nM	[114]
chitosan	click chemistry (PEG)	MNPs (Fe_3_O_4_ core)	optical (colorimetric biosensing assay)	acid-fast bacilli of *M. tuberculosis*	10^2^ CFU·mL^−1^	[113]

**Table 4 nanomaterials-10-01406-t004:** Carbon nanomaterial-based glycan biosensors and their characteristics.

Carbohydrate	Immobilization	Interface	Transducer	Analyte	LOD	Ref.
thiomannosyl dimer	Au–S bond	Multi-walled carbon nanotube (MWCNT)/Au NPs	GCE electrode	lung cells	10 cells·mL^−1^	[124]
				liver cells	40 cells·mL^−1^	
				prostate cells	15 cells·mL^−1^	
mannose	CuAAC chemistry	Borromean rings	mechanical (QCM)	Con A	K_D_ = 147 nM	[115]
		Dodecaamine cages			K_D_ = 1.21 μM	
		fullerenes			K_D_ = 1.02 μM	
glucose, mannose, galactose, maltose and lactose	amide-coupling reaction	CDs	optical	undefined	undefined	[116]
lactose	click chemistry (PEG)			HeLa cells		[117]
				MDA cells		
mannose	amide-coupling reaction	SWCNTs	ELLA (96-well plate)	Con A		[118,119]
		micelles				
	CuAAC chemistry	Borromean rings	mechanical (QCM)		152 nM	[120]
		dodecaamine cages			3.68 μM	
		monovalent reference MR·1 M			K_D_ = 10.9 mM	
Tn antigen	amide-coupling reaction	BSA	GPOx SPE	GOD3-2C4 antibody	10 aM	[121]
				DBA	1 aM	
mannose	click chemistry (PEG)	anthraquinone	GPOx SPE	Con A	down to nM	[123]
				macrophage M2	undefined	
				*E. coli* MG1655	undefined	

**Table 5 nanomaterials-10-01406-t005:** Nanoscale-based platforms for construction of nanobiosensors and their characteristics.

Carbohydrate	Immobilization	Interface	Transducer	Analyte	LOD	Ref.
lactose, galactose and mannose	amide-coupling reaction	hydrogels	optical (2D photonic crystals)	ricin	75 nM	[125]
				jacalin	230 nM	
				Con A	38 nM	
pyrenyl glycoside	click chemistry (PEG)	MoS_2_	electrochemical (graphene SPE–DPV)		373 nM	[126]
				human hepatoma cancer cells	840 cells·mL^−1^	
α-2,6-sialyllactose		MoS_2_ nanosheets	optical	influenza strain H1N1	0.8 HAU·mL^−1^	[127]
				influenza strain H7N9	1.4 HAU·mL^−1^	
α-2,3-sialyllactose						
				influenza strain H10N8	0.4 HAU·mL^−1^	

**Table 6 nanomaterials-10-01406-t006:** Protein-based glycan nanobiosensors and their characteristics.

Carbohydrate	Immobilization	Interface	Transducer	Analyte	LOD	Ref.
mannose	squaric acid conjugation	gold + BSA	optical (SPR)	Con A	1.8 nM	[133,135]
Tn antigen	amide-coupling reaction	polycrystalline Au electrode	electrochemical (EIS)	GOD3-2C4 antibody	270 aM	[134]
		polycrystalline Au electrode + HSA			1.4 aM	
40 glycan epitopes based on *N*-acetyllactosamine	squaric acid conjugation	BSA	microtiter plate	bacterial enterotoxins toxin A and B of *C. difficile*	down to nM	[137]

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
