# Peer review of "Glycan Nanobiosensors"

_nanomaterials, 2020, doi:10.3390/nano10071406_

Round 1
Reviewer 1 Report
This is a timely and thorough collection of references and knowledge in this growing field of measuring technology.
What made it very hard for me to read was the language.
Why do authors so often refrain from professional language editing after having invested so much work in collecting data, writing, drawing figures etc.
THUS, my suggestion actually can also be seen as "Major revision" (but without changes of content)
Author Response
This is a timely and thorough collection of references and knowledge in this growing field of measuring technology.
What made it very hard for me to read was the language.
Why do authors so often refrain from professional language editing after having invested so much work in collecting data, writing, drawing figures etc.
THUS, my suggestion actually can also be seen as "Major revision" (but without changes of content)
R: An extensive English revision of the manuscript was performed by a native English speaker. Please see a revised version of the manuscript with changes highlighted.
Reviewer 2 Report
I quite like the idea of this manuscript, which focuses on a review of glycan biosensors using nanomaterials. Nonetheless, there are a few minor issues (and some relatively less minor ones) that need to be addressed before I can recommend publication.
The minor issues include:
- There are some syntax-based errors that detract from being able to focus exclusively on the scientific content of the manuscript. A detailed proofreading that focuses on these issues is recommended.
- In the introduction, the authors state that “oligosaccharides are able to bind stronger with different glycan-binding proteins than monosaccharides.” This statement is too general – while it is true the vast majority of the time, it is certainly not true all of the time. More clarification should be included.
- Also in the introduction, the authors write that “the process is not a template-driven neither directly encoded in the genome.” This is confusing. The authors should clarify how the process is driven, not merely how it is NOT driven.
- Also in the introduction, the authors indicate that “the role of glycans becomes progressively complex as the structure of glycans itself.” This is a confusing sentence structure and it is unclear what the authors are trying to communicate. More clarification should be provided.
- The authors indicate that “carbohydrate interactions are generally observed between interfaces of larger entities.” While this is true, carbohydrate interactions are certainly not exclusive to these larger entities (implied by the current text). The authors should modify the text to make it clear that carbohydrate interactions due in fact occur in broader contexts.
- The authors write that “biological and medical problems could be solved by glyconanotechnology.” This is an oversimplification. Some of them can be helped by glyconanotechnology; fewer of them can be solved. More tempering of this statement should be included.
- The authors write that “today’s applied toxicity assays are the same for NPs as for classical agents.” They should include a reference to support this assertion. As far as I understand the literature, this statement is inaccurate and there are in fact several nanoparticle-specific toxicity assays that exist. See for example,
- Nguyen, Minh Kim; Moon, Ju-Young; Lee, Young-Chul. Ecotoxicol. Environ. Safety 2020, 201, 110781.
- Savage, Dustin T.; Hilt, J. Zach; Dziubla, Thomas D. Methods in Molecular Biol. 2019, 1894, 1-29.
- Adewale, Olusola B.; Davids, Hajierah; Cairncross, Lynn; Roux, Saartjie. Int. J. Toxicol. 2019, 38, 357-384.
The authors should clarify their statement and/or explain how the aforementioned references fit into their review paradigm.
- The authors indicate that “toxicity corresponds to oxidative stress but not to cellular uptake.” This sentence requires clarification. Toxicity in general in cellular environments requires uptake into cells in order for the mechanism of toxicity to operate. More information should be provided.
- The authors refer to “Vis light” in the text of their manuscript. They should write out the phrase “visible light.”
- The authors also refer to a “MXene 2D nanomaterial.” This phrase, and this class of nanomaterials, needs to be defined and put into a broader context of the types of nanomaterials that exist and are available for glycan sensing applications (see below).
- In general, when the authors discussing the switching platforms around cyclodextrins, they should include an overview of what the benefits are of a switchable sensor, what are challenges that are unique to switchable sensors, and what are the particular advantages/disadvantages of the switchable sensors that they are summarizing herein.
- Immediately after the discussion of the switchable sensors, the authors start talking about printable polymer nanolayers. Again, more background information about this class of materials would be beneficial, as well as information about how nanolayers relate to nanoparticles and the broader class of nanomaterials.
- In the discussion of gold nanoparticles, the authors should make clear from the outset what size nanoparticles they are talking about and how those differ from nanodots/ nanocrystals/ mesoparticles, etc.
- In the same section, the authors indicate that the color of the gold nanoparticles depends on their size. This is a general phenomenon for nanoparticles and one that should be discussed earlier in the review article under a category of “size-dependent optical properties.”
Beyond the minor issues detailed above, this review article appears to be missing some key information at the beginning of the article. In particular,
- This review would strongly benefit from a clear definition of “nanomaterials” and “nanoparticles” at the very beginning of the review, as well as a brief overview of other classes of nanomaterials besides nanoparticles and how “nanoparticles” fits into that broader framework.
- Moreover, in the same part of the review in which an overview of nanomaterials is provided, the authors should provide an overview of the relative advantages/disadvantages of various classes of nanomaterials, including nanoparticles, with a particular role towards sensing and biological applications.
- Finally, in the same section, the authors should indicate how nanoparticles are generally functionalized, what challenges exist with their functionalization, and in what ways functionalization of nanoparticles is preferable to and/or more challenging than functionalizing small molecules.
Author Response
I quite like the idea of this manuscript, which focuses on a review of glycan biosensors using nanomaterials. Nonetheless, there are a few minor issues (and some relatively less minor ones) that need to be addressed before I can recommend publication.
The minor issues include:
- There are some syntax-based errors that detract from being able to focus exclusively on the scientific content of the manuscript. A detailed proofreading that focuses on these issues is recommended.
R: An extensive English revision of the manuscript was performed by a native English speaker. Please see a revised version of the manuscript with changes highlighted.
- In the introduction, the authors state that “oligosaccharides are able to bind stronger with different glycan-binding proteins than monosaccharides.” This statement is too general – while it is true the vast majority of the time, it is certainly not true all of the time. More clarification should be included.
R: A relevant part of the introduction was extended and can be read as follows: “Oligosaccharides are able to bind more strongly with different glycan-binding proteins than monosaccharides, thanks to the variety in structures and conformations [6]. For example lectins bind to monosaccharides with affinity constant in the millimolar range [1]. Oligosaccharides bind to lectins with affinity constant in the micromolar range despite the fact that oligosaccharides can bind to lectins via multiple contacts [1]. This is due to absence of a deeper binding pocket on the surface of lectins allowing competitive solvent interactions [1]. On the other hand, interaction of lectins with monosaccharides can be enhanced in cases lectins are assembled from homo-oligomeric subunits and each subunit interacts with a monosaccharide [1].”
- Also in the introduction, the authors write that “the process is not a template-driven neither directly encoded in the genome.” This is confusing. The authors should clarify how the process is driven, not merely how it is NOT driven.
R: The relevant part was rewritten and can be read as follows: “Glycosylation takes place in the endoplasmic reticulum but more predominantly in the Golgi apparatus and the process is not template-driven. Instead, glycans are synthesised by the action of a series of enzymes [1].”
- Also in the introduction, the authors indicate that “the role of glycans becomes progressively complex as the structure of glycans itself.” This is a confusing sentence structure and it is unclear what the authors are trying to communicate. More clarification should be provided.
R: Relevant part of the text was rewritten and can be read as follows: “For many years, the role of sugars was considered to be primarily nutritional, but the role of glycans becomes progressively more complex [9]. Conjugates are part of interfacial layers of cells and responsible for mediation of the first contact in the host-pathogen interactions [10, 11]. Highly specific but weak protein-glycan interactions are commonly observed in nature and have an essential role in many cellular mechanisms, e.g. cell-cell and cell-biomolecule interactions, stabilisation of tertiary structure of proteins, mechanism of signalling molecules or disease progression and infection of pathogens including toxins, bacteria and viruses [8, 10-13]. As an illustration, sialic acid (SA) is used as a viral receptor molecule by a host cell. Binding of a particular virus to the cell surface is based on interactions between the viral glycoprotein - hemagglutinin (HA) and cell-surface glycans - sialic acid (SA) terminated. Ordinarily, human viral strains are linked to α-2,6-SA moieties, whereas avian viruses predominantly bind to α-2,3-SA structures [14] (Fig. 1).”
- The authors indicate that “carbohydrate interactions are generally observed between interfaces of larger entities.” While this is true, carbohydrate interactions are certainly not exclusive to these larger entities (implied by the current text). The authors should modify the text to make it clear that carbohydrate interactions due in fact occur in broader contexts.
R: This sentence was deleted.
- The authors write that “biological and medical problems could be solved by glyconanotechnology.” This is an oversimplification. Some of them can be helped by glyconanotechnology; fewer of them can be solved. More tempering of this statement should be included.
R: This sentence was rewritten as follows: “Some of the biological and medical problems could be solved by the aid of glyconanotechnology.”
- The authors write that “today’s applied toxicity assays are the same for NPs as for classical agents.” They should include a reference to support this assertion. As far as I understand the literature, this statement is inaccurate and there are in fact several nanoparticle-specific toxicity assays that exist. See for example,
- Nguyen, Minh Kim; Moon, Ju-Young; Lee, Young-Chul. Ecotoxicol. Environ. Safety 2020, 201, 110781.
- Savage, Dustin T.; Hilt, J. Zach; Dziubla, Thomas D. Methods in Molecular Biol. 2019, 1894, 1-29.
- Adewale, Olusola B.; Davids, Hajierah; Cairncross, Lynn; Roux, Saartjie. Int. J. Toxicol. 2019, 38, 357-384.
The authors should clarify their statement and/or explain how the aforementioned references fit into their review paradigm.
- The authors indicate that “toxicity corresponds to oxidative stress but not to cellular uptake.” This sentence requires clarification. Toxicity in general in cellular environments requires uptake into cells in order for the mechanism of toxicity to operate. More information should be provided.
R: The relevant part of the manuscript was rewritten taking into account Q7 and Q8 and all three references (Ref. 30-32) suggested by the Reviewer. The updated text can be read as follows: “It is challenging to compare the toxicity of nanomaterials with their macroscale counterparts. Current toxicity assays are applied equally both to NPs and to conventional agents. Hence, the evaluation of NPs’ toxicity by current methods may not be sufficient. This is due to increased surface area of NPs strongly adsorbing active agents applied for traditional toxicology assays; optical properties of NPs, which might interfere with fluorescence or optical detection systems; and magnetic properties of NPs, which might interfere with methods based on redox reactions [30]. As a result of such properties of NPs, the methods used for traditional toxicology studies cannot be directly applied for examination of toxicity of NPs [30]. Moreover, NPs can be accumulated in various organs for longer periods than traditional pharmaceutical agents generating oxidative stress, inflammation, cell death and agglomerates within vessels [31]. There is a need for the development of new assays consisting of several approaches [31] including assessment of toxicity of NPs by aquatic organisms [32], which will take into consideration factors such as size, shape, surface area, surface charge, porosity or hydrophobicity which influence the functions and toxicity of nanomaterials [33, 34]. It has been established that NPs are able to activate an immune response and induce phagocytic cells that will eliminate them from the bloodstream, or may induce immunostimulation which may promote inflammatory disorders, or even immunosuppression which increases the host’s susceptibility to infections and cancer.”
- The authors refer to “Vis light” in the text of their manuscript. They should write out the phrase “visible light.”
R: This was corrected.
- The authors also refer to a “MXene 2D nanomaterial.” This phrase, and this class of nanomaterials, needs to be defined and put into a broader context of the types of nanomaterials that exist and are available for glycan sensing applications (see below).
R: A term MXene is now shortly explained by the following text: “An alternative to covalent grafting of aryl diazonium species is a spontaneous grafting in case a nanomaterial contains a free electron cloud (plasmons) as in the case of hybrid magnetic particles with a gold shell [49, 50] or MXene 2D nanomaterial (a novel form of a 2D hydrophilic nanomaterial made of alternating layers of two elements such as titanium and carbon i.e. Ti3C2) [51].”
- In general, when the authors discussing the switching platforms around cyclodextrins, they should include an overview of what the benefits are of a switchable sensor, what are challenges that are unique to switchable sensors, and what are the particular advantages/disadvantages of the switchable sensors that they are summarizing herein.
R: A relevant part of the text was rewritten as follows: “However, some immobilisation strategies are included here since they facilitate the preparation of switchable interfaces. The beauty of switching approaches described below is regeneration of the interfaces by external trigger in this particular case by light. More information about switchable materials can be found elsewhere [44].”
- Immediately after the discussion of the switchable sensors, the authors start talking about printable polymer nanolayers. Again, more background information about this class of materials would be beneficial, as well as information about how nanolayers relate to nanoparticles and the broader class of nanomaterials.
R: The sentence was completed and can be read as follows: “Upon laser irradiation, polymer-embedded molecules are released and can be deposited on another surface with the thickness of printed nanolayers of 10 nm.”
- In the discussion of gold nanoparticles, the authors should make clear from the outset what size nanoparticles they are talking about and how those differ from nanodots/ nanocrystals/ mesoparticles, etc.
R: The following text was added into the text of the manuscript: “In the following section we are discussing AuNPs with the size above the size of nanodots i.e. 10 nm [72] and mainly without nanotextured surface, which is typical for mesoparticles [73].”
- In the same section, the authors indicate that the color of the gold nanoparticles depends on their size. This is a general phenomenon for nanoparticles and one that should be discussed earlier in the review article under a category of “size-dependent optical properties.”
R: Relevant part of the manuscript was updated and can be read as follows: “The aggregation of AuNPs results in a pronounced colour change from red to blue (shift to longer wavelengths) dependent on inter-particle distances (Fig. 6) [63]. Colour changes of NPs are closely correlated with the resonance between an oscillation of electrons and the incident electromagnetic radiation [74]. It is worth mentioning that optical properties are changed with size for other types of NPs such as quantum dots [75], as well.“
Beyond the minor issues detailed above, this review article appears to be missing some key information at the beginning of the article. In particular,
- This review would strongly benefit from a clear definition of “nanomaterials” and “nanoparticles” at the very beginning of the review, as well as a brief overview of other classes of nanomaterials besides nanoparticles and how “nanoparticles” fits into that broader framework.
R: The following sentence was added into Section 2: “Here we use a term nanomaterial as a material, which consists of structured components (nanoparticles) with at least one dimension less than 100 nm [21].”
- Moreover, in the same part of the review in which an overview of nanomaterials is provided, the authors should provide an overview of the relative advantages/disadvantages of various classes of nanomaterials, including nanoparticles, with a particular role towards sensing and biological applications.
- Finally, in the same section, the authors should indicate how nanoparticles are generally functionalized, what challenges exist with their functionalization, and in what ways functionalization of nanoparticles is preferable to and/or more challenging than functionalizing small molecules.
R: The following text was added to address these two comments in the revised version of the manuscript: “AuNPs are quite frequently applied for construction of biosensors since such nanoparticles allow modifying interface in a convenient way using formation of self-assembled monolayers (SAMs) using thiolated biomolecules [43]. A combination of two thiols can effectively tune density of functional groups deposited on the surface and glycan nanobiosensors on AuNPs can be directly formed using thiolated glycans [44]. Although formation of SAM on AuNPs is straightforward with a lot of beneficial properties, it is very important to point out to the fact that components attached to AuNPs via SAM are highly mobile with a possibility that glycan will form clusters over time even though glycans were initially homogeneously distributed over the interface [44]. AuNPs are frequently applied to design electrochemical (modification of the electrodes) and optical (signal nanoprobes) biosensor devices. Quantum dots (QDs), which are usually terminated in hydroxyl groups can be quite easily modified by silane chemistry with final formation of a strong irreversible bond. The initial step is hydrolysis of silane with subsequent condensation reaction creating Si-O bond [44]. Similar to thiolated SAMs, silane chemistry allows forming a monolayer with a possibility to deliver various functional groups, but the process is more difficult to control [44]. QDs are especially applied obviously in combination with optical detection, but also electrochemical detection platform is frequently applied since heavy metals of QDs can be effectively detected by electrochemical means. QDs are mainly applied as signal nanoprobes. Magnetic particles are usually made of Fe3O4 meaning that interfacial hydroxyl group can be again applied for modification via silane chemistry discussed above for QDs. The obvious advantage of using magnetic particles is to apply them for preconcentration/enrichment of the analyte from complex samples [45]. Magnetic particles can be loaded with various labels to form signal nanoprobes. Carbon nanoparticles can be modified via various routes. For example planar forms of carbon nanoparticles such graphene or carbon nanotubes can be modified by non-covalent p- p stacking interactions, by covalent grafting of molecules to modified/oxidised carbon nanoparticles or via electrochemically triggered grafting of molecules having diazonium moieties. Carbon nanoparticles are frequently applied for modification of the electrodes or as signal nanoprobes for electrochemical biosensing or can be applied for fluorescent biosensing with carbon nanoparticles effectively applied for fluorescence quenching. Bioconjugation protocols applicable for glycan immobilisation are in details discussed in our book chapter [44].”